# Bile Acids: Major Regulator of the Gut Microbiome

**DOI:** 10.3390/microorganisms10091792

**Published:** 2022-09-06

**Authors:** Chihyeok An, Hyeyeon Chon, Wanrim Ku, Sunho Eom, Mingyu Seok, Sangha Kim, Jaesun Lee, Daesung Kim, Sanghyuk Lee, Hoonsup Koo, Hyunjung Cho, Seungyun Han, Juik Moon, Miil Kang, Kihyun Ryu

**Affiliations:** 1Department of Gastroenterology, Konyang University Myunggok Medical Research Institute, Daejeon 35365, Korea; 2Department of Laboratory Medicine, Konyang University College of Medicine, Daejeon 35365, Korea; 3Department of Anatomy, Konyang University College of Medicine, Daejeon 35365, Korea; 4Department of Surgery, Konyang University College of Medicine, Daejeon 35365, Korea; 5Department of Rheumatology and Clinical Immunology, Dankook University Hospital, Cheonan 31116, Korea

**Keywords:** bile acids, gut microbiome, enterohepatic circulation

## Abstract

Bile acids are synthesized from cholesterol and play an important role in regulating intestinal microflora. The different degrees of hydrophobicity and acidity of individual bile acids may affect their antimicrobial properties. We examined the antimicrobial effects of different bile acids on various microorganisms in vitro and confirmed whether these remain consistent in vivo. Using human bile acids, including ursodeoxycholic acid, cholic acid, chenodeoxycholic acid, deoxycholic acid, and lithocholic acid, a disc diffusion test was performed, and a rodent model was created to determine the antimicrobial effects of each bile acid. The fecal bacterial population was analyzed using a real-time polymerase chain reaction. Each bile acid showed different microbial inhibitory properties. The inhibitory activity of bile acids against microbiota which normally resides in the gastrointestinal tract and biliary system, was low; however, normal flora of other organs was significantly inhibited. Changes in microbial counts after bile acid administration in a rodent model differed in the colon and cecum. The in vivo and in vitro results show that the antimicrobial effects of bile acids against intestinal microbiota were similar. In conclusion, bile acids could be a novel treatment strategy to regulate gut microbiota.

## 1. Introduction

The bile acid pool, formed from the enterohepatic circulation, is one of the major regulators of the intestinal microbiota [1]. Bile acids, synthesized from cholesterol in the liver, are converted into cholic acid (CA) through the classic pathway and chenodeoxycholic acid (CDCA) through the alternative pathway. These primary bile acids are secreted into the intestinal tract and dehydroxylated by the intestinal microbiota to form secondary bile acids, which are deoxycholic acid (DCA) and lithocholic acid (LCA). Only 5% of bile acid is excreted in the stool through the colon, while the rest is reabsorbed in the small intestine [2]. Therefore, the liver produces only 500–600 mg of bile acid per day, which is approximately 5% of the total bile acid volume [3]. The reabsorbed bile acids are conjugated in the liver and secreted into the intestinal tract in various forms based on their conjugation status. Bile acids exhibit various degrees of hydrophobicity and acidity; thus, various bile acids differentially affect intestinal microbiota [3,4].

Bile acids exhibit bacteriostatic and bactericidal effects against microbiota by disrupting their membranes [5,6], denaturing proteins [7,8], chelating iron and calcium [9,10], damaging DNA [5,11], and regulating host immunity via nuclear receptors such as farnesoid X receptor (FXR; NR1H4) and vitamin D receptor (VDR; NR1I1) [12,13]. However, microorganisms possess diverse defense mechanisms, such as cell envelope remodeling [14,15], modulation of the efflux system [16,17], DNA repair [14,15], and stress response [18,19], to counter the antimicrobial effects of bile acids. Moreover, intestinal microorganisms influence the composition of the intestinal bile acid pool by converting primary bile acid into secondary bile acid or by deconjugation, epimerization, and dehydroxylation [20]. Consequently, the interaction between bile acids and intestinal microbiota contributes in maintaining intestinal homeostasis.

The proportion of specific bile acids gradually increases when they are supplied into the gastrointestinal tract through continuous enterohepatic circulation and become the major component of the bile acid pool. Thus, supplementing some bile acids, such as ursodeoxycholic acid (UDCA), can be a potential therapeutic strategy. UDCA accounts for less than 5% of the total bile acid volume; continuous oral administration can increase its proportion to up to 50% of the total bile acid volume [21]. Thus, continuous supplementation of certain bile acids not only affects the bile acid pool of the intestinal tract but can also change the composition of the intestinal microbiota.

The large intestine comprises the cecum, colon, rectum, and anal canal [22]. The cecum, the first portion of the large intestine, is a large blind pouch below the ileocecal valve [23]. In humans, the cecum plays a minor role in intestinal physiology. However, the rodent cecum, which is largely developed, is where food is kneaded and digested. The cecum is a site for functional fermentation as it contains a high concentration of microorganisms and functions as a reservoir of microorganisms supplied to the colon [24]. Studies on the intestinal microbiota in rodent models are mostly based on stool samples from the cecum. In contrast to humans, the rodent cecum lumen is clearly separated from the colon lumen. Bile from the ileocecal valve flows through the colon, rectum, and anal canal; however, bile flows inside the cecum, which is located on the opposite side [25]. Thus, the rodent cecal microbiota may not be an appropriate animal model to represent human colonic microbiota.

In this study, we examined the antimicrobial effects of different bile acids on various microorganisms in vitro and by performing the disc diffusion test, which is used to determine the antibiotic susceptibility of microorganisms. Additionally, we investigated the antimicrobial effects of the bile acid pool that was enriched with specific bile acids in vivo using a rodent model. We intended to conduct in vivo experiments with strains commonly found in the biliary tract and intestinal tract. Among them, *Escherichia coli*, *Proteus mirabilis*, and *Enterococcus faecalis*, which are known to selectively inhabit the colon and cecum, and those in our test target organs, were selected [26]. A standard strain capable of mass culture was selected and purchased. In addition, we added *Lactobacillus casei*, which is the most popular commercially available probiotic, has undergone many studies, and is easy to culture. Fecal samples from the colon and cecum of mice were collected to evaluate the differences in the microbiota composition.

## 2. Materials and Methods

### 2.1. Inhibition of Bacteria by Bile Acid

#### 2.1.1. Strains and Growth Conditions

The susceptibility of the following bacterial strains detected in the biliary or intestinal tract to bile acids was evaluated: *Clostridium difficile* (ATCC 43598), *Enterobacter cloacae* (ATCC 700323), *Enterococcus casseliflavus* (ATCC 700327), *Escherichia coli* (ATCC 25922), *Klebsiella oxytoca* (ATCC 700324), *Klebsiella oxytoca* (ATCC 13182), *Klebsiella pneumoniae* (ATCC 700503), and *Proteus mirabilis* (ATCC 12453). Additionally, the susceptibility of the following bacterial strains that cause opportunistic infections in organs other than the intestine was analyzed: *Serratia marcescens* (ATCC 14756), *Staphylococcus aureus* (ATCC 29243), *Staphylococcus epidermidis* (ATCC 14966), *Staphylococcus xylosus* (ATCC 29971), *Streptococcus pneumoniae* (ATCC 700503), and *Streptococcus pyogenes* (ATCC 19615). The susceptibility of *Candida albicans* (ATCC 10231), one of the most common fungi in the human gut flora, to bile acids was also analyzed.

*Enterobacter cloacae, Escherichia coli, Klebsiella oxytoca, Klebsiella pneumoniae*, and *Proteus mirabilis* were subcultured in Asan MAC I (Asan Pharm. Co., Ltd., Gyeonggi-do, Korea) for 24 h at 37 °C under static conditions. *Clostridium difficile* was cultured on CDIF agar medium (Asan Pharm. Co., Ltd.) for 24 h at 37 °C in an anaerobic chamber. The cultures were stored at 4 °C between transfers and were subcultured once overnight in a fresh culture medium before use. *Enterococcus casseliflavus*, *Enterococcus faecalis*, *Pseudomonas aureginosa*, *Staphylococcus aureus*, *Staphylococcus epidermidis*, *Staphylococcus xylosus*, *Streptococcus pneumoniae*, *Streptococcus pyogenes*, and *Candida albicans* were subcultured in Asan BAP II (Asan Pharm. Co., Ltd.) for 24 h at 37 °C under static conditions.

Additionally, we analyzed the susceptibility of the following commercially available probiotics to bile acids: *Clostridium butyricum*, *Bacillus mesentericus* (Biotap Cap., Daewoong Pharm, Seoul, Korea), *Lactobacillus casei* (Ramnos Granule; Hanhwa Pharm Co., Ltd., Seoul, Korea), and *Saccharomyces boulardii* (fungusi Bioflor 250 Powder; Kunil Pharm, Seoul, Korea). These probiotics were maintained by subculturing 1% inoculum in MRS broth (Becton and Dickinson Company, Sparks, MD, USA) for 24 h at 37 °C under static conditions.

#### 2.1.2. Bile Acid Susceptibility Test

CA, LCA, CDCA, DCA, and UDCA were purchased from Sigma–Aldrich (St. Louis, MO, USA). Each bile acid (0.2 g) was dissolved in 1 mL solvent. Next, 20 μL of bile acid solution was placed on a sterile 5-mm blank disc and allowed to dry at room temperature for about 10 min; this process was repeated thrice. The discs were subjected to UV sterilization for 30 min on a clean bench. The dried disc was stored at –20 °C until use. The disc diffusion test currently recommended by the NCCIS subcommittee on antimicrobial susceptibility testing is a slight modification of the protocol described by Bauer, Kirby, Sherris, and Turck [27]. The isolated colonies of each microorganism were mixed in sterile 1-mL normal saline and then streaked onto the dried surface of the Mueller–Hinton agar plate (Asan Pharm. Co., Ltd.). The discs inoculated with bile acids were placed on the surface of the bacterium-inoculated agar plate and gently pressed on the agar surface using sterile forceps to ensure complete contact. The discs were placed on the plates individually such that the discs were no closer than 15 mm from the edge of the plate and no two discs were closer than 24 mm (center to center) from each other. The plates were then inverted and placed in a 37 °C incubator for 24–48 h. The diameter of the zone of complete inhibition (as judged by the unaided eye) was measured to the nearest whole millimeter using a ruler prepared for this purpose. The measurements were performed by placing the ruler on the back of the petri plate illuminated with reflected light. Unlike antibiotic susceptibility testing, the inhibition range was quantified by subtracting the disc diameter from the total diameter to clarify the non-inhibiting bile acids.

### 2.2. Bile Acid Administration to the Mouse Model

#### 2.2.1. Animals

All mouse experiments were approved by the Institutional Animal Care and Use Committee (IACUC; approved protocol number: 18-15-A-01) of the Konyang University (Daejeon, Korea). Six-week-old male C57BL/6 mice (n = 18, bodyweight 18–22 g) were purchased from SAMTAKO (SAMTAKO Co., Ltd., Gyeonggi-do, Korea). All mice used in the experiments were housed in groups of 3 per cage under the same conditions. The mice were acclimated in an environmentally controlled room at 23 ± 2 °C with 40 ± 10% relative humidity in a 12-h light/dark cycle. All animals had free access to water and food. All experiments were conducted in accordance with the “Guide for the Care and Use of Laboratory Animals” (National Institutes of Health publication 8th edition, 2011).

#### 2.2.2. Experimental Design

The mice were randomly divided into the following groups (n = 3 per group):

Group I (Vehicle), treated only with solvent; Group II, 15 mg/kg CA treatment (CA); Group III, 15 mg/kg LCA treatment (LCA); Group IV, 15 mg/kg DCA treatment (DCA); Group V, 15 mg/kg CDCA treatment (CDCA); and Group VI, 15 mg/kg UDCA treatment (UDCA). The bile acids were dissolved in sodium bicarbonate solution (20 mg/mL). The bile acid or solvent (vehicle group) was intragastrically administered for 21 days. The body weight of the animals was measured every day. On day 21 after treatment, all mice were weighed and sacrificed using 1 ug/mL urethane (0.5 mL/100 g bodyweight).

#### 2.2.3. Stool Collection and Real-Time Polymerase Chain Reaction (qRT-PCR)

Feces were collected every week during the period of bile acid administration. Intraluminal stools in the cecum and colon were collected on the day of the sacrifice.

After sacrifice, the large intestine was separated by double ligation using a silk thread at the end of the terminal ileum, cecal inlet, and proximal rectum. Subsequently, the double-ligated thread was cut. The cecum and colon were completely separated, and feces in each luminal space were collected. The fecal samples were homogenized manually. The cecal or colonic bacterial population was analyzed by qRT-PCR. DNA was isolated from the luminal contents of the ligated segments of the cecum or colon using the QIAamp Fast Stool DNA Extraction kit (Qiagen, Hildon, Germany), following the manufacturer’s instructions. The total bacterial levels were quantified by qRT-PCR using the universal bacterial-specific primers, 5′-AAACTCAAAKGAATTGACGG-3′ (For) and 5′-CTCACRRCACGAGCTGAC-3′ (Rev) in a Lightcycler^®^ 96 (Roche, Basel, Switzerland). The following primers were used to quantify the bacteria at the phylum level: *Bacteroidetes*, 5′-GTTTAATTCGATGATACGCGAG-3′ (For) and 5′-TTAASCCGACACCTCACGG-3′ (Rev); *Firmicutes*, 5′-GGAGYATGTGGTTTAATTCGAAGCA-3′ (For) and 5′-AGCTGACGACAACCATGCAC-3′ (Rev). The following primer pairs were used to quantify the bacteria at the species level: *Escherichia coli*, 5′-GAAGCTTGCTCTTTGCTGA-3′ (For) and 5′-CTTTGGTCTTGCGACGTTAT-3′ (Rev); *Enterococcus faecalis*, 5′-GGGGACAGTTTTGGATGCTA-3′ (For) and 5′-TCCATATAGGCTTGGGCAAC-3′ (Rev); *Lactobacillus casei*, 5′-AGCAGTAGGGAATCTTCCA-3′ (For) and 5′-CACCGCTACACATGGAG-3′ (Rev).

The qRT-PCR was performed in a 20-μL reaction volume containing 10 μL of BioFACTTM 2X Real-Time PCR Master MIX, dNTP(T) and SFCgreen^®^ mixture, Low ROX (BIOFACT Co., Ltd., Daejeon, Korea), 3 μL of ultra-pure water (Welgene Precision SolutionTM Daegu, Korea), 1 μL of 10 pmol each of the forward and reverse primers, and 5 μL of cecal or rectal bacterial DNA samples. The PCR conditions were as follows: 95 °C for 5 min, followed by 45 cycles of 95 °C for 10 s, 55 °C for 20 s, and 72 °C for 30 s. For *Lactobacillus*, the following PCR conditions were used: 95 °C for 5 min, followed by 45 cycles of 95 °C for 10 s, 50 °C for 20 s, and 72 °C for 30 s. The fluorescent products were detected at the last step of each cycle. The melting curve analysis was performed after amplification to distinguish the targeted PCR product from the non-targeted PCR product.

All target bacterial DNA fragments were analyzed to evaluate the change in abundance of the bacterial population. Quantification of the target microbial strains in each mouse was calculated by relative quantitative analysis based on the cycle quantification value (Cq) of the universal bacteria primer of vehicle group # 1.

#### 2.2.4. Statistical Analysis

Continuous variables are presented as the mean ± SD. One-way analysis of variance (ANOVA) with the Tukey honestly significant difference (HSD) test was applied to determine whether there were any statistically significant differences between the mean of 6 groups. The student’s *t*-test was used to compare the mean values of the colon and cecum. All statistical analyses were performed using SPSS Statistics for Windows (v. 27, IBM Corp., Armonk, NY, USA).

## 3. Results

### 3.1. Inhibition of Bacteria by Bile Acid

The most hydrophilic and hydrophobic bile acids are shown in the left and right bars of Figure 1A–C, respectively. The susceptibility of microorganisms to the bile acids was determined based on the inhibition zone diameter in the disc diffusion test.

The bile acids exhibited differential antimicrobial effects against the microorganisms.

Treatment with bile acids resulted in a clear inhibition zone in the strains found outside the intestine (Figure 1A), with CDCA and DCA exhibiting the highest inhibition zone diameter. However, the inhibition zone diameter varied with each bile acid, and none of the bile acids exhibited a consistent inhibition pattern. Additionally, there was no linearly proportional relationship between the antimicrobial activity and hydrophobicity of bile acid. Although LCA was strongly hydrophobic, it did not exhibit antimicrobial activity against any strains.

The bile acids exhibited minimal antimicrobial activity (narrow inhibition zone) against the microorganisms mainly detected in the intestine and biliary system (Figure 1B). DCA and CDCA exhibited strong antimicrobial activity, but the inhibition zone diameter varied according to the species. Compared to the strains found outside the intestine, the sensitivity of strains found inside the intestine to bile acids was low. LCA did not exhibit antimicrobial activity against most strains. In *Klebsiella oxytoca* and *Clostridium difficile*, narrow inhibition zones were observed.

Many probiotic microbes were inhibited by bile acids, especially by CA, CDCA, and DCA (Figure 1C).

### 3.2. Bile Acid Injection Model

The body weights (Figure 2) of mice in the DCA, CDCA, and UDCA treatment groups were non-significantly lower than those in the vehicle, CA, and LCA treatment groups.

The levels of glutamate pyruvate transaminase and total bilirubin in the CDCA group were non-significantly higher than in the other groups. (Table 1) Other test values also showed no significant difference.

### 3.3. Microbiota in Colon and Cecum

The continuous supplementation of a bile acid into the gut changed the intestinal microbiota composition in the animal model (Figure 3).

Statistically, the difference in population numbers between groups according to the administration of various bile acids was significant in the cecum (*p* < 0.05) but not in the colon. (Appendix A) The number of microbes in the colon and cecum increased in the UDCA treatment group. Overall, the number of microorganisms in the colon was higher than that of the cecum, but it was not statistically significant compared to each group. Changes in microbial counts with bile acid administration were observed inversely in the colon and cecum except in the UDCA treatment group. (See the curve above the bar in Figure 3). CDCA and DCA exhibited higher antimicrobial activity against colonic microbes than LCA and UDCA. The levels of colonic microbes in the CDCA and DCA treatment groups and the levels of cecal microbes in the CA and LCA treatment groups were similar to those in the vehicle treatment group. The CDCA and DCA treatment groups exhibited an enhanced number of microbes in the cecum. The Firmicutes/Bacteroidetes ratio (F/B ratio) in the colon decreased, whereas that in the cecum increased after treatment with bile acids. This reverse phenomenon of the F/B ratio was statistically significant in the remaining four groups except for the vehicle-administered group and the LCA-administered group (*p* < 0.05) (Appendix A).

### 3.4. Comparative Analysis of In Vitro and In Vivo Analysis

We comparatively evaluated the in vitro and in vivo antimicrobial effects of bile acids against three intestinal microbes (*Escherichia coli*, *Proteus mirabilis*, and *Enterococcus faecalis*) frequently observed in the human large intestine and most popular probiotics (*Lactobacillus casei*) (Figure 4). Consistent with the disc diffusion test results, the number of specific microbiotas in the colon changed after the mice were treated with each bile acid. However, the number of cecal microbes was unaffected by bile acid treatment.

## 4. Discussion

Bile acids can affect the microbial composition of the intestinal tract through bacteriostatic and bactericidal effects. The intestinal microbiota maintains the homeostasis of bile acids in the body by converting primary bile acids into secondary bile acids or deconjugation [20]. Bile acids may exhibit direct antimicrobial effects through their detergent action [4] or indirect inhibitory effects through the FXR and VDR [28]. The antimicrobial effects of different bile acids vary depending on the degree of hydrophobicity and affinity to FXR and VDR. Additionally, different microorganisms exhibit varying degrees of susceptibility to different bile acids [29]. This study aimed to evaluate whether the gut microbiome is affected by artificial compositional changes in bile acids through in vitro and in vivo experiments.

As shown in Figure 1, the microorganisms detected in the intestine were less susceptible to bile acids when compared to the pathogens or opportunistic infectious microorganisms, which are rarely detected in the intestinal tract and commonly detected in other organs. It is suggested that there is something in the intestine that inhibits the growth of certain strains. The levels of *Klebsiella pneumoniae* and *Enterococcus faecalis*, which are resistant to bile acids, increase significantly in the biliary system during infections. This indicated that bile acids play an important role in inhibiting certain microorganisms [30].

We hypothesized that the hydrophobic characteristic of bile acids is responsible for their antimicrobial effects through detergent action; however, this study revealed that the antimicrobial effects of bile acids are not correlated with hydrophobicity. Although LCA is a highly hydrophobic bile acid, it exhibited low antimicrobial activity in vitro. The antimicrobial activity varied widely across different microorganisms, even among similar bile acid types. As the disc diffusion test is an in vitro experiment, the interaction of bile acids with the FXR and VDR, intranuclear bile acid receptors, cannot be evaluated. Thus, in addition to the hydrophobic characteristics, other factors may contribute to the antimicrobial activity of bile acids. Additionally, microorganisms can affect bile acid composition, which was not evaluated in this study.

Previous studies have demonstrated that bile acids are important regulators of intestinal microorganisms [5,6,7,8,9,10,11,12,13,14,15,16,17,18,19]. In the real world, the intestinal microbial composition was mostly regulated by externally supplying the microorganisms(probiotics). As shown in Figure 1C, most bile acids exhibited strong antimicrobial activity against commercially available probiotics, suggesting that bile acids may inhibit externally supplemented probiotics. Therefore, the intestinal microbial composition will eventually return to the previous state observed before probiotics supplementation.

Bile acids can be supplemented in drug form and as a single or combined form. The principal bile acids present in human beings are less likely to be rejected by the host, and most side effects are predictable. Additionally, bile acid supplementation from humans minimizes side effects. The concentration of a target bile acid can be detected by fecal sampling, and the treatment progression can be monitored. Additionally, the amount of bile acid supplementation can be decreased or even terminated when side effects are observed. Thus, supplementation of bile acids is an attractive strategy.

In the animal study model, the cecum and colon were expected to exhibit differential microbial compositions (Figure 3). Consistent with this hypothesis, the colonic microbial composition in the CDCA and DCA treatment groups was like that in the control group. However, the cecal microbial composition in the CDCA and DCA treatment groups was different from that in the control group. Interestingly, the *Firmicutes*/*Bacteroidetes* ratio was also reversed, suggesting that the cecum may act as a reservoir of colonic microbiotas and that microorganisms migrate from the cecum to the colon to compensate for changes in the colon. Thus, the rest of the colon, except the cecum, will be suitable for studying the human gastrointestinal tract.

However, the animal model used in this study has many limitations; the results obtained cannot be directly extrapolated to humans. Additionally, the bile acids inhibit the microorganisms and affect the host metabolism. The interaction between gut microbiota, host factors, and bile acids results in the enrichment of the specific bile acid.

The mammalian intestinal environment varies in the small intestine, cecum, and large intestine. The physiological environments vary due to nutrient concentrations and host immune reaction differences. The small intestine has a lower pH and higher oxygen concentration than the large intestine. Additionally, the small intestine contains a large volume of bile acids and antimicrobial substances that inhibit the growth of microorganisms. As the small intestine is the main organ that absorbs nutrients, the growth of microorganisms that compete for nutrients is suppressed. However, the microorganisms in the small intestine and fast-growing facultative anaerobes, such as *Lactobacillacea* and *Enterobacteriacea,* which consume simple carbohydrates, are resistant to bile acids. Meanwhile, the cecum and colon are known to have the most diverse microbes in the human body. The function of the cecum is believed to not be important in humans.; however, herbivorous animals and rodents have a relatively large cecum in the transitional region of the ileum and colon. Plant fiber is known to be digested by microbes, such as *Ruminococcaceae* and *Lachnospiraceae*. The microbes in the colon digest the polysaccharides that are not absorbed by the small intestine. Thus, fermentative polysaccharide-degrading anaerobes, such as *Clostridia* and *Bacteroidacea,* are abundant in the large intestine due to the slow migration of digestive products, low concentration of antimicrobial substances, and absence of monosaccharides [31]. This study also revealed the differential effects of bile acids on the colonic and cecal microbial composition in rodents.

The composition and density of microbes, which are immunologically regulated via FXR and VDR, may be highly diverse in the intrafold region and luminal center as well as in the intestinal mucosal layer and epithelium. In the human intestine, a mucus layer formed from goblet cells forms a boundary between the intestinal lumen and soft tissue. The small intestine is covered with a single mucus layer, while the large intestine is covered with two mucus layers. In the large intestine, the inner mucus layer is tightly associated with the intestine. Meanwhile, the outer mucus layer is loosely associated with the intestine, is dense, and harbors several microbes. In contrast, the inner mucus layer is almost devoid of microbes. The internal mucosal layer is in direct contact with the colonic epithelium, which aids in maintaining high concentrations of oxygen and antimicrobial substances secreted from the epithelial cells [32]. The microbial composition of the colonic mucosal layer is different from that of the lumen. The mucus layer contains many mucin-degradable microorganisms, such as *Bacteroides acidifaciens*, *Bacteroides fragilis*, and *Akkermansia muciniphila* belonging to *Antinobacteria* and *Proteobacteria* phyla [33,34]. This study only detected bacteria in the feces of the intestinal lumen and did not evaluate the effects of bile acids on the microorganisms in the mucus layer. Thus, the effects of bile acids on the human intestinal microbiota may provide varied results depending on the site and method of detection.

The composition of bile acids in rodents is different from that in humans. The major bile acids in humans are CA and CDCA, whereas those in rodents are CA and β-muricholic acid (β-MCA) [35]. The rodent intestinal bile acid content also varies, with hyodeoxycholic acid (HDCA) being the most abundant bile acid in the cecum (41%) [36]. HDCA has a hydrophilic-hydrophobic index (HHI) of +0.09, which is closer to human CA (HHI: +0.23) and has a higher hydrophobicity than UDCA (HHI: −0.17) [37]. Additionally, there are small amounts of bile acids, such as β-hyocholic acid (β-HCA), α-muricholic acid (α-MCA), β-MCA, and murideoxycholic acid (MDCA), which are hardly detected in humans.

There were some technical limitations in this study. The appearance and thinness of the stool varied depending on the bowel environment of the individual. According to the observed results, the number of strains in the cecum was relatively stable within the same group. In contrast, in the colon, there was a large difference related to the condition of each individual. This pattern was also observed in other experiments in our group that analyzed the number of microorganisms in the colon. This could be overcome in experiments with a large number of subjects. In this study, we tried to rectify this inaccuracy by dividing the relative number of microbes by the weight of the feces. The stool weight was calculated by comparing the weight before and after the stool was added to the buffer. However, the difference in the water content of feces was not corrected because microbes are also present in the liquid part of the stool. Analyzing the microbes only in the solid part (by centrifugation) may be misleading. Additionally, the number of anaerobic microbes in the stool may change after exposure to air. To compensate, the stool was directly placed into the buffer to minimize contact with air.

Moreover, the total numbers of *Bacteroidetes* and *Firmicutes* exceeded the total number of microbes (Figure 4). This indicates the limitation of using primers because some microorganisms often have one or more identical DNA sequences. Future studies must focus on addressing this limitation.

## 5. Conclusions

Bile acids exhibit marked antimicrobial effects against intestinal microbiota depending on the type of microbial strain and specific bile acid. The antimicrobial effects of bile acids against intestinal microbiota were similar in vitro and in vivo. The results of this study indicate that colonic microbes are more susceptible to bile acids than cecal microbes.

The administration of bile acids, probiotics, and fecal transplantation may be a potential strategy to regulate intestinal microbiota. Additionally, the rodent colonic microbiota rather than the cecal microbiota may more accurately represent the human colonic microbiota.

## Figures and Tables

**Figure 1 microorganisms-10-01792-f001:**
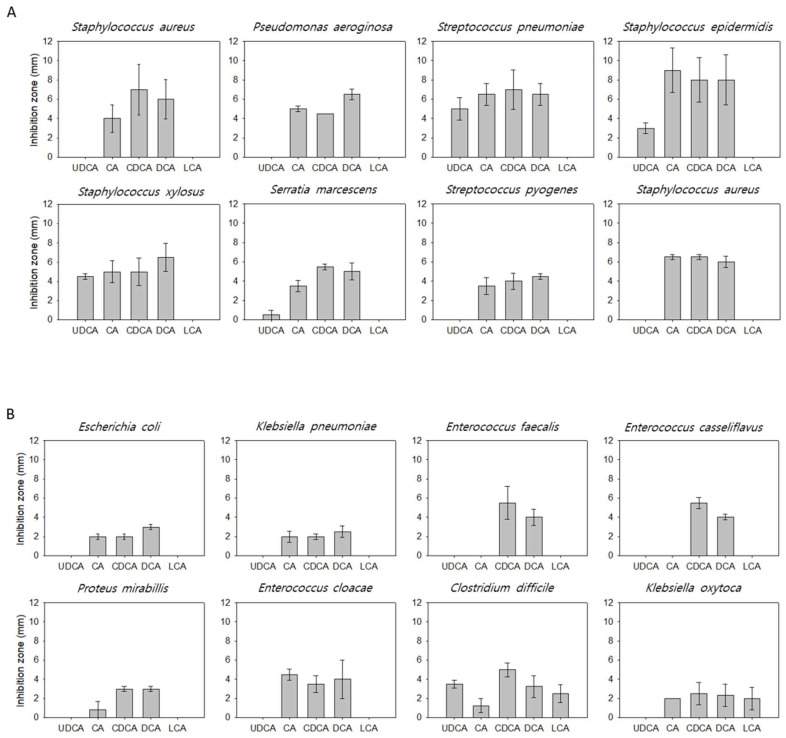
(**A**). Susceptibility of strains found outside the intestine to bile acids. (**B**). Susceptibility of strains mainly observed in the intestine and biliary system to bile acids. (**C**). Susceptibility of commercially available probiotics to bile acids.

**Figure 2 microorganisms-10-01792-f002:**
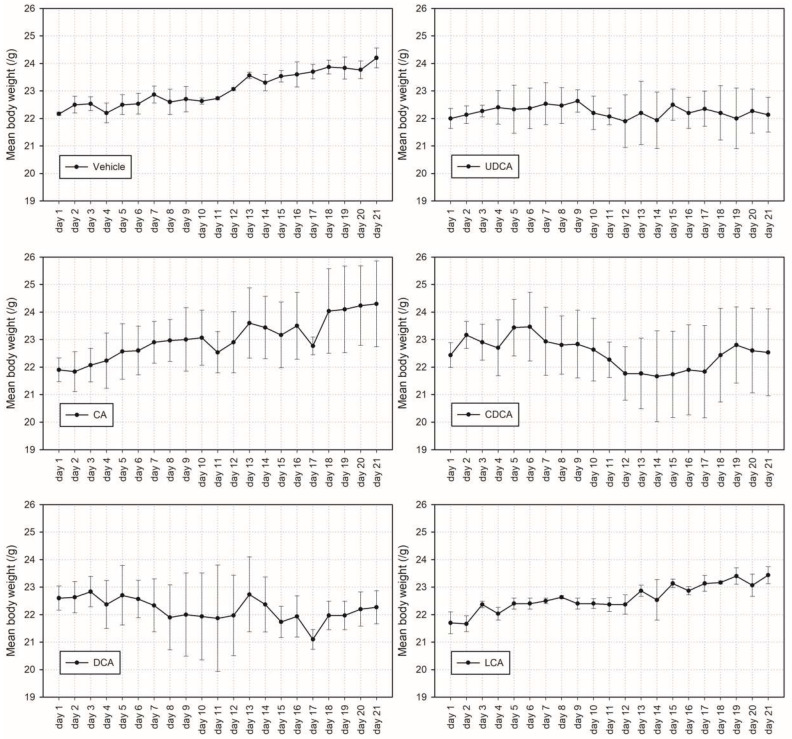
Average bodyweight of mice treated with different bile acids.

**Figure 3 microorganisms-10-01792-f003:**
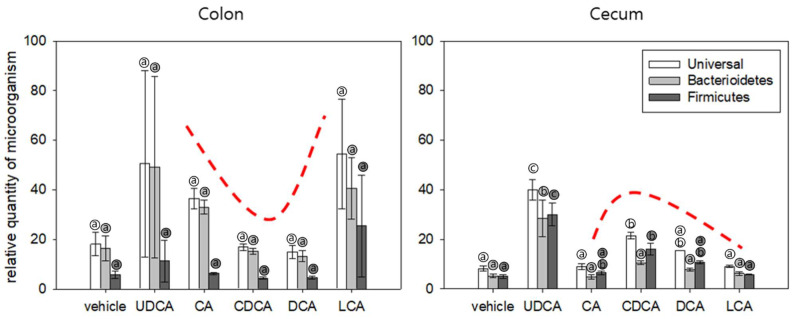
Microbiotas in the colon and cecum. The graph on the left represents the gut microbiota obtained from the colon when the mice were sacrificed, and the graph on the right represents the gut microbiota from the cecum. The horizontal axis indicates the group treated with each bile acid, and the vertical axis indicates the relative quantity of microorganisms. White bars represent quantification results using the universal bacterial primer for the genome sequence common to all bacteria, gray represents *Bacteriodetes*, and black represents *Firmicutes*. The same letters in the small shaded circle indicate a non-significant difference between groups, and significant differences were indicated with different letters. Each phylum was shaded the same as in the graph bars. Changes in microbial counts with bile acid administration were observed inversely in the colon and cecum except in the UDCA treatment group. (See the curve above the bar).

**Figure 4 microorganisms-10-01792-f004:**
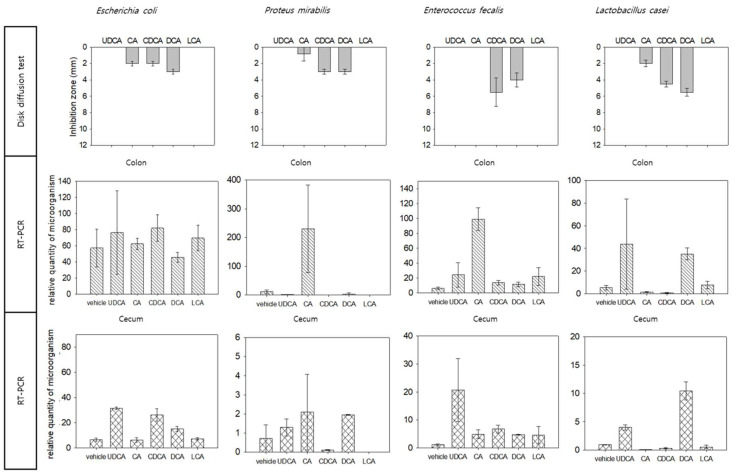
Comparative analysis of in vitro and in vivo results. Direct inhibition of bile acids against specific selected strains (in vitro) and the effect on mice large intestine strains (in vivo) were compared. For in vitro experiments, disc diffusion was used, and quantitative analysis (RT-PCR) of strains in the large intestine was performed separately in the colon and cecum.

**Table 1 microorganisms-10-01792-t001:** Results of analysis of blood samples taken during sacrifice.

	Vehicle	CA	LCA	DCA	CDCA	UDCA
Glucose, mg/dL	334.7 ± 16.2	262.3 ± 13.0	268.3 ± 46.9	212.3 ± 32.9	214.0 ± 28.4	245.7 ± 17.7
Total cholesterol, mg/dL	103.3 ± 6.2	114.7 ± 6.1	105.3 ± 13.0	107.5 ± 14.7	108.3 ± 11.1	119.3 ± 12.5
GPT, U/L	7.7 ± 0.3	8.0 ± 1.0	6.7 ± 0.9	6.0 ± 0.6	12.3 ± 4.1	6.0 ± 0.6
T.bil, mg/dL	1.9 ± 0.3	1.7 ± 0.5	1.7 ± 0.1	1.2 ± 0.2	3.2 ± 0.8	1.6 ± 0.0
ALP, U/L	124.7 ± 4.5	116.7 ± 15.9	164.3 ± 14.5	187.0 ± 1.7	116.7 ± 34.1	129.3 ± 33.1
GGT, U/L	6.7 ± 5.2	5.0 ± 2.6	1.0 ± 0.0	1.0 ± 0.2	31.3 ± 18.9	5.3 ± 4.3
TG, mg/dL	65.3 ± 16.2	58.0 ± 7.8	39.0 ± 3.0	35.0 ± 1.2	44.0 ± 8.0	33.7 ± 6.0
Albumin, g/dL	2.5 ± 0.1	2.3 ± 0.1	2.4 ± 0.1	2.3 ± 0.2	2.4 ± 0.2	2.1 ± 0.1
HDLC, mg/dl	89.5 ± 0.9	96.0 ± 2.5	96.7 ± 6.9	106.0 ± 2.3	104.0 ± 4.0	96.7 ± 6.8
GOT, U/L	312.5 ± 43.6	376.0 ± 100.7	250.3 ± 69.7	142.3 ± 1.8	146.0 ± 2.3	227.0 ± 1.2
Total protein, g/dL	5.0 ± 0.9	5.1 ± 0.1	4.9 ± 0.8	5.3 ± 0.4	5.4 ± 0.2	4.7 ± 0.2

Data are expressed in mean ± standard deviation. GPT, glutamate pyruvate transaminase; T.bil, total bilirubin; ALP, alkaline phosphatase; GGT, gamma-glutamyltransferase; TG, triglyceride; HDLC, high-density lipoprotein cholesterol; GOT, glutamic oxaloacetic transaminase; UDCA, ursodeoxycholic acid; CA, cholic acid; CDCA, chenodeoxycholic acid; DCA, deoxycholic acid; LCA, lithocholic acid.

## Data Availability

Data sharing not applicable.

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
