# Peer review of "Bile Acids: Major Regulator of the Gut Microbiome"

_microorganisms, 2022, doi:10.3390/microorganisms10091792_

Round 1

Reviewer 1 Report

Overall, it is an interesting study. The research content is also suitable for publication in the Journal of Microorganisms. My specific comments are as follows:

1.       The reference is not in the correct format.

2.       Introduction: there are too many descriptions, and the focus and significance of the research content are not well expressed. For example, the second and fourth paragraphs can be combined and the logical order can be adjusted; In addition, the author needs to explain why several bacteria are selected in the in vitro bacteriostatic experiment.

3.       Table 1: Why did the author not perform significance analysis?

4.       Discussion: the author needs to discuss the results of the study and delete unnecessary explanations.

5.       This will need to undergo review by a language editor to correct some issues with grammar and syntax.

Reviewer 2 Report

In this manuscript, the authors present a study on the microbial regulatory properties of bile acids. The manuscript is well structured and methodologies are appropriate. However, there are serious flaws in the introduction and discussion regarding the structure of the gut microbiota and its regulation. There are many sentences that state facts that are not correct and are very well studied in many articles in reviews. Additionally, these stated facts are not suported by any reference, which suggests that there has not been conducted a proper reading on the area, since there is a lot of literature on it.  I would recommend acceptance following a deep revision of the introduction and discussion, especially the last one. The lab work is fine and interesting, it´s the context what states ideas that are misleading. Comments below:

- In vitro and in vivo should go in italics.

-Line 35. The bile acid pool is not the major regulator of the intestinal microbiota at all. It is the diet, among other factors. You can say that "...circulation, is, arguably, one of the major regulators of the intestinal microbiota". There is plenty of literature, please add also a reference. 

-Line 49. Add a reference at the end of that sentence.

-Line 50 and 309. There are no 1000 phyla in the human gut microbiome. If you think there are, please cite the reference. 

-Line 54. Bile acids can exhibit bacteriostatic and bactericidal effect and microorganisms can exhibit defense mechanisms, but you are making absolute statements without providing references. Not all microorganisms are affected by bile acids, but some of them are. References.

-Line 65-67. This is not right, intestinal microbiome is not restored by external supplementation. I guess what you mean is that the human microbiome is resilient, but this is not just because of the bile acid composition. Again, if you are making such strong statements, cite where you took the idea from. 

-Figure 2. It is difficult to appreciate the different treatments, could you rearrange the figure so they can be appreciated?

-Table 1. Title should be self-explanatory. Elaborate a bit more. 

-Figure 3. Elaborate the legend a bit more. Where are the statistics that prove that you can make the statement that microbes are increasing or decreasing? p-values? The error bars in the UDCA and LCA are huge, you have to discuss that, bar errors that high give no meaning to the result.

-Figure 4. Elaborate the legend a bit more. 

-Lines 276-278. This is a wrong statement. Gut microbiota composition is characteristic of each individual, but it is not only determined by the bile acid composition. Add the reference. 

Overall, you need more references for your statements. Lower the tone of the claims and add references. 

Round 2

Reviewer 1 Report

At present, this manuscript can be accepted for publication.

Author Response

Thank you for your review of our artwork.

We contacted a proofreading company and did a little more English proofreading.

-Line 289

• “The antimicrobial effects of different bile acids varies depending on the degree of hydrophobicity and affinity to FXR and VDR.” have been revised as “The antimicrobial effects of different bile acids vary depending on the degree of hydrophobicity and affinity to FXR and VDR.”

-Line 300

• “This indicated that bile acids may plays an important role in inhibiting certain microorganisms.” have been revised as “This indicated that bile acids play an important role in inhibiting certain microorganisms.”

Reviewer 2 Report

I would like to thank the authors for going thoroughly through all the comments and taking the time to address them and explain them. I think the manuscript has been improved and I will be happy to accept it after just three points that I think need to be addressed before final publishing. I am sorry, I do not want to be the difficult reviewer, but it is important that there are not things that can induce confusion. 

1. Please add this reference at the end of the first sentence of your introduction (line 37). https://www.ncbi.nlm.nih.gov/pmc/articles/PMC4215539/

2. (This is just a comment). Don´t worry about the phyla issue, that can happen.

3. If you have added statistics to your text (as you have done), please add a subsection (a couple of lines) in the methods section, explaining which statistics you have used and add a sign into the figures and tables where it is significant. 

4. Figure 2 is still confusing, it´s not about removing the error bars for clarity, you can just do different smaller pannels for each treatment with the vehicle and the error bars, so it is easier to appreciate everything.

Author Response

Response to Reviewer’s Comments

Sep 1, 2022

Prof. Dr. Martin Von Bergen

Editor-in-Chief

microorganisms

Department of Molecular Systems Biology

Helmholtz Centre for Environmental Research

Leipzig, Germany

Dear Prof. Dr. Martin Von Bergen:

We would like to thank you and the reviewers of microorganisms for taking the time to review our article twice. We have made some corrections and clarifications in the manuscript (Manuscript ID: microorganisms-1865360) after going over the reviewers’ comments. The changes are summarized below:

* Reviewer #2

I would like to thank the authors for going thoroughly through all the comments and taking the time to address them and explain them. I think the manuscript has been improved and I will be happy to accept it after just three points that I think need to be addressed before final publishing. I am sorry, I do not want to be the difficult reviewer, but it is important that there are not things that can induce confusion.

Comment 1:

Please add this reference at the end of the first sentence of your introduction (line 37). https://www.ncbi.nlm.nih.gov/pmc/articles/PMC4215539/

Our response: We have added that reference.

Comment 2:

(This is just a comment). Don´t worry about the phyla issue, that can happen.

Our response: Thank you for your generosity.

Comment 3:

If you have added statistics to your text (as you have done), please add a subsection (a couple of lines) in the methods section, explaining which statistics you have used and add a sign into the figures and tables where it is significant.

Our response: Thanks for telling us what we overlooked.

  1. We added the following to the methods section:

2.2.4. Statistical analysis

Continuous variables are presented as the mean ± SD. One-way analysis of variance (ANOVA) with the Tukey honestly significant difference (HSD) test was applied to determine whether there are any statistically significant differences between the mean of 6 groups. Student's t-test was used to compare the mean values of colon and cecum. All statistical analyses were performed using SPSS Statistics for Windows (v. 27, IBM Corp. Armonk, NY, USA).

  1. Since the presentation of the analysis results is complicated, we expressed it as follows.

2-1. We marked significant differences with different letters in figure 3 (If there is no significant differences between groups they get the same letter). Each phylum was shaded the same as in the graph.

[Before change]

Figure 3. Microbiotas in the colon and cecum. The graph on the left represents the gut microbiota obtained from the colon when the mice were sacrificed, and the graph on the right represents the gut microbiota from the cecum. The horizontal axis indicates the group treated with each bile acid, and the vertical axis indicates the relative quantify of microorganisms. White bars represent quantification results using the universal bacterial primer for the genome sequence common to all bacteria, gray represents Bacteriodetes, and black represents Firmicutes.

[After change]

Figure 3. Microbiotas in the colon and cecum. The graph on the left represents the gut microbiota obtained from the colon when the mice were sacrificed, and the graph on the right represents the gut microbiota from the cecum. The horizontal axis indicates the group treated with each bile acid, and the vertical axis indicates the relative quantify of microorganisms. White bars represent quantification results using the universal bacterial primer for the genome sequence common to all bacteria, gray represents Bacteriodetes, and black represents Firmicutes. The same letters in the shaded small circle indicate non-significant difference between groups and significant differences were indicated with different letters. Each phylum was shaded the same as in the graph bars.

2-2. We provided data tables as supplementary materials.

Supplementary Table 1. Comparison between groups of relative quantity of microbiotas in colon and cecum.

  1. Colon

Relative

 quantity

Groups

Significance value*

vehicle

UDCA

CA

CDCA

DCA

LCA

Universal

T

22.90±18.07

a

22.94±9.36

a

23.75±11.35

a

18.80±8.56

a

60.25±57.84

a

42.78±48.04

a

NS

(P= .596)

Bacteriotetes

T

16.35±8.71

a

49.13±63.58

a

33.02±4.81

a

15.16±2.16

a

13.28±3.76

a

40.65±21.56

a

NS

(P= .515)

Firmicutes

T

5.78± 2.47

a

11.33±14.44

a

6.32±0.88

a

4.42±0.84

a

4.66±1.04

a

25.42±35.34

a

NS

(P= .564)

  1. Cecum

Relative

 quantity

Groups

Significance value*

vehicle

UDCA

CA

CDCA

DCA

LCA

Universal

T

8.21±1.73

a

39.93±7.27

c

8.94±1.99

a

21.45±2.31

b

15.47±0.16

a,b

9.01±0.74

a

P= .000

Bacteriotetes

T

5.26±1.07

a

28.38±12.71

b

4.80±1.53

a

10.58±1.47

a

7.91±1.01

a

6.21±1.14

a

P= .001

Firmicutes

T

5.08±1.22

a

29.98±7.83

c

6.57±1.27

a,b

15.99±4.13

b

10.75±0.93

a,b

5.80±0.31

a

P= .000

* Statistical significances were tested by oneway analysis of variances among group.

† The same letters indicate non-significant difference between groups based on Tukey’s multiple comparison test.

Supplementary Table 2. Comparison of Firmicutes/Bacteroidetes ratio (F/B ratio) between colon and cecum.

Relative quantity

of microbiota

Groups

vehicle

UDCA

CA

CDCA

DCA

LCA

Universal

Colon

22.90±18.07

22.94±9.36

23.74±11.35

18.80±8.56

60.25±57.84

42.78±48.04

Cecum

8.21±1.73

39.93±7.27

8.94±1.99

21.45±2.31

15.47±0.16

9.00±0.74

Bacteriotetes

Colon

16.35±8.71

49.13±63.58

33.02±4.81*

15.16±2.16*

13.28±3.76

40.65±21.56

Cecum

5.26±1.07

28.38±12.71

4.80±1.53*

10.58±1.47*

7.90±1.01

6.21±1.14

Firmicutes

Colon

5.78±2.47

11.32±14.44

6.32±0.88

4.42±0.84*

4.66±1.04*

25.42±35.34

Cecum

5.08±1.22

29.98±7.83

6.57±1.27

15.99±4.13*

10.75±0.93*

5.80±0.31

F/B ratio

Colon

3.17±1.73

4.25±0.17*

5.37±1.53*

3.45±0.17*

2.86±0.55*

4.26±2.95

Cecum

1.11±0.49

0.91±0.19*

0.73±0.17*

0.68±0.19*

0.73±0.03*

1.08±0.23

* The mean difference is significant at the 0.05 level (p< .05; t-test) between colon and cecum.

Comment 4:

Figure 2 is still confusing; it´s not about removing the error bars for clarity, you can just do different smaller panels for each treatment with the vehicle and the error bars, so it is easier to appreciate everything.

Our response: Thanks for the good suggestion. We modified it as follows according to the method you suggested.

[Before change]

[After change]

We hope the revised manuscript will better meet the requirements of your journal for publication. We thank the editor and the reviewers of microorganisms once again for the constructive review of our paper.

Sincerely yours,

Kihyun Ryu, M.D., Ph.D.

Associate professor

Division of Gastroenterology and Hepatology

Konyang University College of Medicine

Daejeon, Korea

Tel: +82-10-7464-0620/ Fax: 82-42-600-9090
